# Convergent shifts in host-associated microbial communities across environmentally elicited phenotypes

Tyler J. Carrier [1] & Adam M. Reitzel[1]

Morphological plasticity is a genotype-by-environment interaction that enables organisms to increase fitness across varying environments. Symbioses with diverse microbiota may aid in acclimating to this variation, but whether the associated bacteria community is phenotype specific remains understudied. Here we induce morphological plasticity in three species of sea urchin larvae and measure changes in the associated bacterial community. While each host species has unique bacterial communities, the expression of plasticity results in the convergence on a phenotype-specific microbiome that is, in part, driven by differential association with α- and γ-proteobacteria. Furthermore, these results suggest that phenotype-specific signatures are the product of the environment and are correlated with ingestive and digestive structures. By manipulating diet quantity over time, we also show that differentially associating with microbiota along a phenotypic continuum is bidirectional. Taken together, our data support the idea of a phenotype-specific microbial community and that phenotypic plasticity extends beyond a genotype-by-environment interaction.

[1] Department of Biological Sciences, University of North Carolina at Charlotte, Charlotte, 28223 NC, USA. Correspondence and requests for materials should be addressed to T.J.C. (email: tcarrie1@uncc.edu)

Phenotypic plasticity, the ability of a single genotype to produce multiple distinct phenotypes, is a genotype-by-environment interaction that enables organisms to acclimate to environmental variation[1–3]. For many organisms, including plants, amphibians, and marine invertebrates, plasticity confers a fitness advantage (i.e., is adaptive) when the phenotype matches the environment[4]. The context-dependent expression of alternate phenotypes is, therefore, presumed to be an evolvable trait influenced by natural selection[4]. To date, ecological and evolutionary theory, including that of phenotypic plasticity, is primarily viewed as a genotype-by-environment interaction[1,2,4–6]. However, all eukaryotes, including plants and animals, are not strictly biological individuals[7] but, instead, are holobionts that comprise a host and consortium of associated microbiota[8–12].

The hologenome theory of evolution proposes that multicellular eukaryotes establish partnerships with microbiota (e.g., eukaryotes, bacteria, Archaea, fungi, and viruses) that are, in part, heritable and affect fitness[8,10–12]. Variation in these microbial communities may arise from differential associations following changes in the environment or host and/or microbial genomes[8]. The hologenome of a holobiont is, therefore, a host genome-by-microbial metagenome-by-environment ($G_H$x$G_M$xE) interaction, whereby acclimation is the result of changes in both $G_M$ and $G_H$[8,13]. Environment-mediated shifts in the structure (i.e., composition and abundance) of host-associated microbial communities often vary in response to biotic challenges, such as diet type[14,15] and starvation[16]. Shifts in host-associated microbial communities may, therefore, co-occur with the expression of environmentally elicited and adaptive morphological characters.

A system to test the hypothesis that host-associated microbial communities are phenotype specific are the feeding (planktotrophic) larvae of marine invertebrates. Planktotrophic larvae require exogenous nutrients to progress through development and undergo metamorphosis[17]. The abundance and distribution of phytoplankton in coastal seas are spatially and temporally heterogeneous and often diluted in offshore waters[18]. Several groups of planktotrophic larvae, including echinoids (phylum Echinodermata, class Echinoidea), respond to heterogeneous feeding environments by exhibiting morphological plasticity[19]. When experiencing starvation, larvae allocate energetic resources from development of the larval body toward the structures for ingestion (i.e., post-oral arms) while absorbing stomach tissues, enabling larvae to increase their feeding capacity in low food environments[20–25]. The role of and responses by the associated microbial community along this morphological continuum remains unknown, even though echinoderm larvae associate with diverse microbial communities[26] and encounter tremendous numbers of environmental microbiota[27].

While within-species comparisons may discern the potential values of phenotypic plasticity, comparisons of conserved responses between closely related species to common environmental variation provides a broader inference for characterizing shared and species-specific adaptive responses. Here we use larvae of three confamilial echinoid species[28] (Strongylocentrotus purpuratus, Mesocentrotus franciscanus, and S. droebachiensis; Supplementary Fig. 1A-C) that differ in their expression of plasticity[29], in order to test the hypothesis that the associated microbial community co-varies with expression and magnitude of morphological plasticity. Through a series of differential feeding experiments paired with sequence-based analysis of the associated bacterial community, we provide evidence that the microbiome shifts following the expression of phenotypic plasticity and the magnitude to which this character is expressed.

## Results

### Larval morphometrics

Morphological plasticity in echinoid larvae is induced following exposure to a low phytoplankton environment. To induce plasticity in S. purpuratus, M. franciscanus, and S. droebachiensis, larvae were fed 10,000; 1000; 100; or 0 cells mL$^{-1}$ of the cryptophyte Rhodomonas lens. As predicted, each species exhibited a significant morphological change upon 4 weeks of differential feeding (Fig. 1; Supplementary Fig. 2; Supplementary Tables 1–3; analysis of variance (ANOVA), $p < 0.001$). Plasticity of the feeding structures was observed for S. purpuratus, M. franciscanus, and S. droebachiensis, where larvae at the same developmental stage fed the same diet exhibited a higher post-oral arm to mid-body line ratio with time (Fig. 1; Supplementary Fig. 2; Supplementary Tables 2, 3; ANOVA, $p < 0.001$). For the time points where plasticity was expressed, the magnitude of morphological change was inversely correlated with the degree of maternal investment (Fig. 1; Supplementary Fig. 3).

For S. purpuratus, larvae fed 100 cells mL$^{-1}$ exhibited morphological plasticity following 2 versus 3 or 4 weeks of diet restriction (Fig. 1a; Supplementary Fig. 2A), where the ratio between post-oral arms and larval body increased, on average, by 10.9% ($\pm$0.8%). For M. franciscanus, plasticity was observed when comparing larvae fed 100 cells mL$^{-1}$ following 1 versus 2 or 3 weeks of diet restriction (Fig. 1b; Supplementary Fig. 2B), where the ratio between post-oral arms and larval body increased, on average, by 9.1% ($\pm$1.4%). Lastly, for S. droebachiensis, larvae fed 1000 cells mL$^{-1}$ expressed plasticity following 2 versus 3 or 4 weeks of diet restriction (Fig. 1c; Supplementary Fig. 2C), where the ratio between post-oral arms and larval body increased, on average, by 4.5% ($\pm$2.1%).

### Microbiome across morphological plasticity states

We used our morphological plasticity data as reference points to compare the structure of the microbiome along this phenotypic transition. The composition of the associated microbial community for larvae of each species was distinct between phenotypes when developmental stage and diet were identical (analysis of similarity (ANOSIM), S. purpuratus: $p < 0.004$, M. franciscanus: $p < 0.006$, S. droebachiensis: $p = 0.046$; Fig. 2a–c; Supplementary Fig. 4–6).

Next, we tested whether the magnitude of morphological change was correlated with the magnitude to which the associated bacterial community was restructured. We determined that the number of differentially associated operational taxonomic units (OTUs) from pre- to post-expression of plasticity was directionally proportional to the magnitude of morphological change ($R^2 = 0.938$; Fig. 3a) and inversely proportional to egg size (Supplementary Fig. 7A). Specifically, S. purpuratus, M. franciscanus, and S. droebachiensis differentially associated with 446, 302, and 152 OTUs (Supplementary Fig. 7), respectively. Furthermore, the ratio between over- to under-represented OTUs was directionally proportional to the magnitude of morphological change ($R^2 = 0.880$; Fig. 3b; Supplementary Fig. 7C) and inversely proportional to egg size (Supplementary Fig. 7B).

Recruitment and expulsion of bacteria and/or a shuffling of relative proportion of the resident communities are not mutually exclusive mechanisms for differentially associating with microbial taxa[8]. By comparing the relative abundance of bacteria at higher taxonomic levels along this phenotypic transition, we observed that larvae trade-off in associating with α- and γ-proteobacteria. Larvae from each species associated with relatively more γ-proteobacteria and less α-proteobacteria following the expression of phenotypic plasticity (Fig. 2d–f). Furthermore, we observed a similar phenotype-specific trade-off at both the family and genus level for each species of larvae. Specifically, the γ-proteobacteria Colwelliaceae, Oleispira, and Pseudomonas for S. purpuratus, Colwelliaceae for M. franciscanus, and Flavobacteriaceae (e.g., Polaribacter) for S. droebachiensis represent a greater portion of the associated microbial communities of larvae

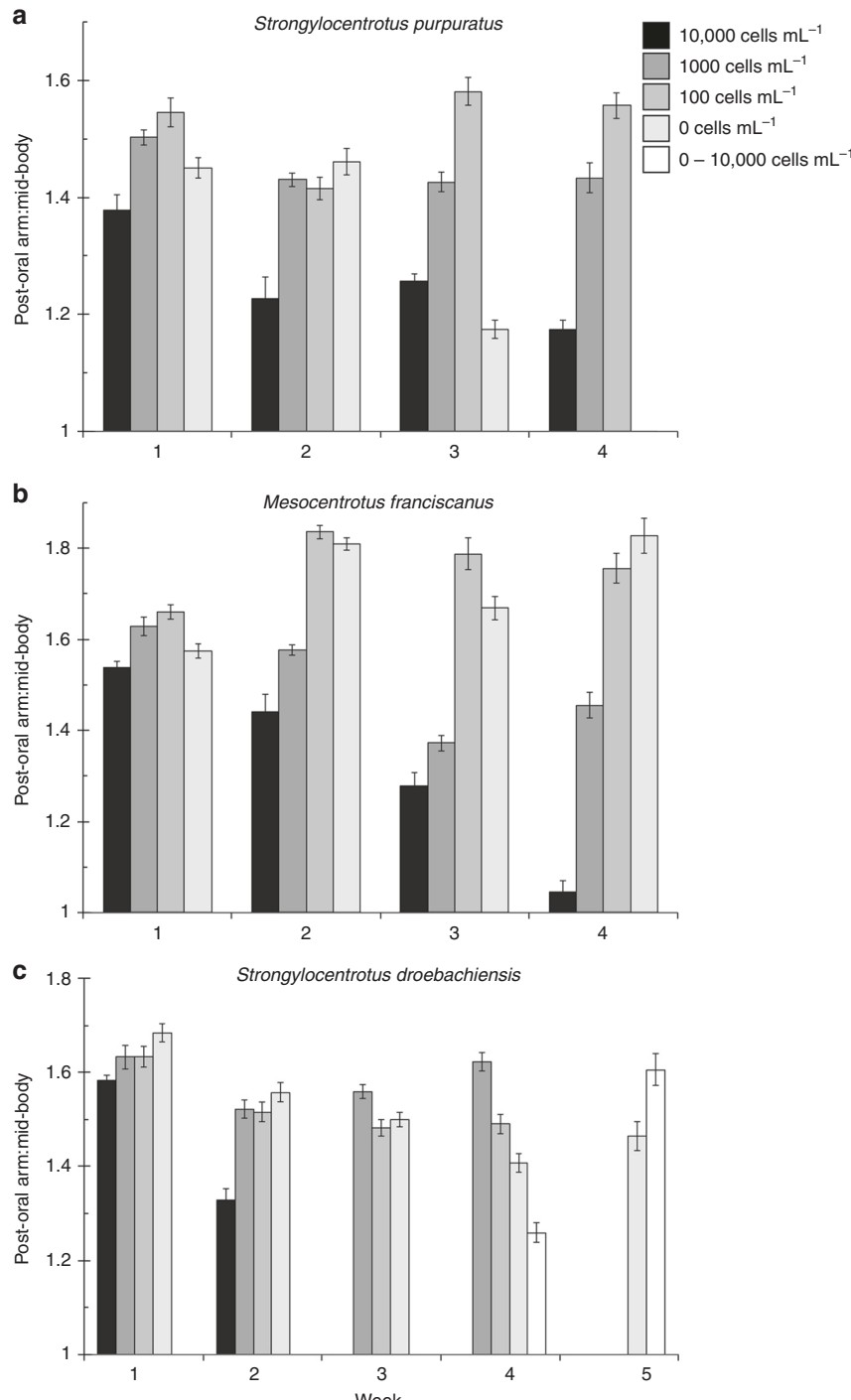

**Fig. 1** Three species of echinoid larvae alter phenotype to feeding environment. Ratio between the post-oral arm and mid-body line (mean ± standard error; $n = 20$; Supplementary Fig. 1D) for *Strongylocentrotus purpuratus* (**a**), *Mesocentrotus franciscanus* (**b**), and *S. droebachiensis* (**c**) larvae having been fed either 10,000 (black), 1000 (dark gray), 100 (gray), and 0 cells mL$^{-1}$ (light gray). For *S. droebachiensis*, larval phenotype was also manipulated (white) by being fed 0 cells mL$^{-1}$ for 3 weeks, then transferred to 10,000 cells mL$^{-1}$ for 3 weeks (i.e., until metamorphosis)

having exhibited phenotypic plasticity (Supplementary Fig. 8; Supplementary Table 4, 5; ANOVA, $p < 0.008$). On the other hand, the α-proteobacteria Bradyrhizobiaceae for *S. purpuratus*, *Sphingomonas* for *M. franciscanus*, and Bradyrhizobiaceae for *S. droebachiensis* represent a reduced portion of the associated microbial communities of larvae having exhibited phenotypic plasticity (Supplementary Fig. 8; Supplementary Table 4, 5; ANOVA, $p < 0.008$).

**Diet- and development-based shifts in the microbiome**. Nutritional developmental plasticity in echinoid larvae is induced when shifted from a well-fed to diet-restricted feeding regime[19,23–25]. To test whether differences in the structure of the associated microbial community were a product of the feeding environment (i.e., quantity of phytoplankton), we compared community similarity across dietary states following 1 week (i.e., pre-expression of plasticity) of

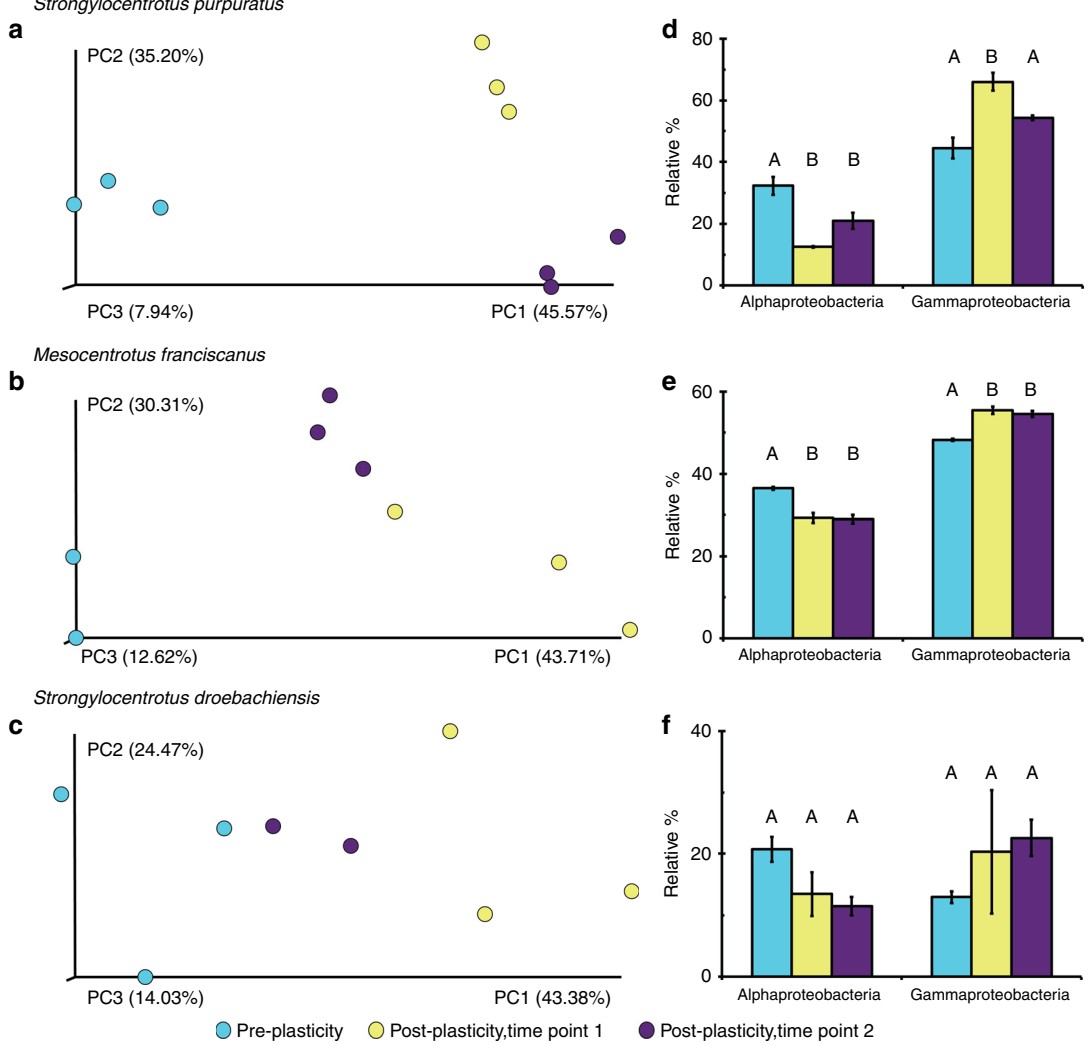

**Fig. 2** Similarity of the associated microbial community along a phenotypic continuum for three species of echinoid larvae. Community similarity of the associated microbiota for *Strongylocentrotus purpuratus* (**a**), *Mesocentrotus franciscanus* (**b**), and *S. droebachiensis* (**c**) prior to (blue) and post (purple and yellow) expression of phenotypic plasticity. Along this continuum, larvae differentially associate with α- and γ-proteobacteria pre- and post-expression of phenotypic plasticity, respectively (**d** *S. purpuratus*; **e** *M. franciscanus*; **f** *S. droebachiensis*)

differential feeding and at a later time point (i.e., post-expression of plasticity).

For *S. purpuratus* and *M. franciscanus*, but not *S. droebachiensis*, the structure of the microbiome was similar across food treatments following 1 week of differential feeding (ANOSIM, *S. purpuratus*: $p = 0.325$, *M. franciscanus*: $p = 0.808$, *S. droebachiensis*: $p < 0.002$; Fig. 4a–c; Supplementary Fig. 9A–C, 10–12). At later time points following the expression of phenotypic plasticity, the structure of the microbiome was distinct across food treatments for all species (ANOSIM, *S. purpuratus*: $p < 0.002$, *M. franciscanus*: $p < 0.003$, *S. droebachiensis*: $p < 0.002$; Fig. 4d–f; Supplementary Fig. 9D–F, 13–15). This difference between weeks pre- and post-expression of plasticity supports that larval phenotype and the associated microbiota was likely the product of differential feeding.

A confounding factor specific to *S. droebachiensis* following one week of differential feeding was that developmental stage was variable across diets, where higher concentrations resulted in advanced stages (Supplementary Table 1). To test whether echinoid larvae associated a developmental stage-specific microbial community (as defined by the number of larval arms), we compared community similarity of 4-, 6-, and 8-arm *S.*

*purpuratus* larvae reared on the same diet and exhibiting a similar plasticity state. Similar to other taxa[30,31], *S. purpuratus* associated with a developmental stage-specific microbial community (ANOSIM, $p < 0.005$; Supplementary Fig. 16). We hypothesize that the difference in associated microbial community observed in *S. droebachiensis* 1 week post differential feeding was, in part, due to a mixed population of 4- and 6-arm larvae across diets (Supplementary Table 1).

**De-coupling phenotypic plasticity.** Each component of nutritional developmental plasticity for echinoid larvae (i.e., diet, development, and phenotype) has specific microbial communities (Figs. 2 and 4; Supplementary Fig. 16). These components, however, are biologically linked and thus our results may, in part, be explained by co-variation between these factors. To de-couple diet, development, phenotype, and time (i.e., ecological drift), we compared the microbial communities of 4-, 6-, and 8-arm larvae of both *S. purpuratus* and *M. franciscanus* fed 100; 1000; and 10,000 cells mL$^{-1}$, respectively, to 4-arm larvae pre- and post-expression of phenotypic plasticity (i.e., larvae from Fig. 2a, b). For both species, we observe a diet–development coupling

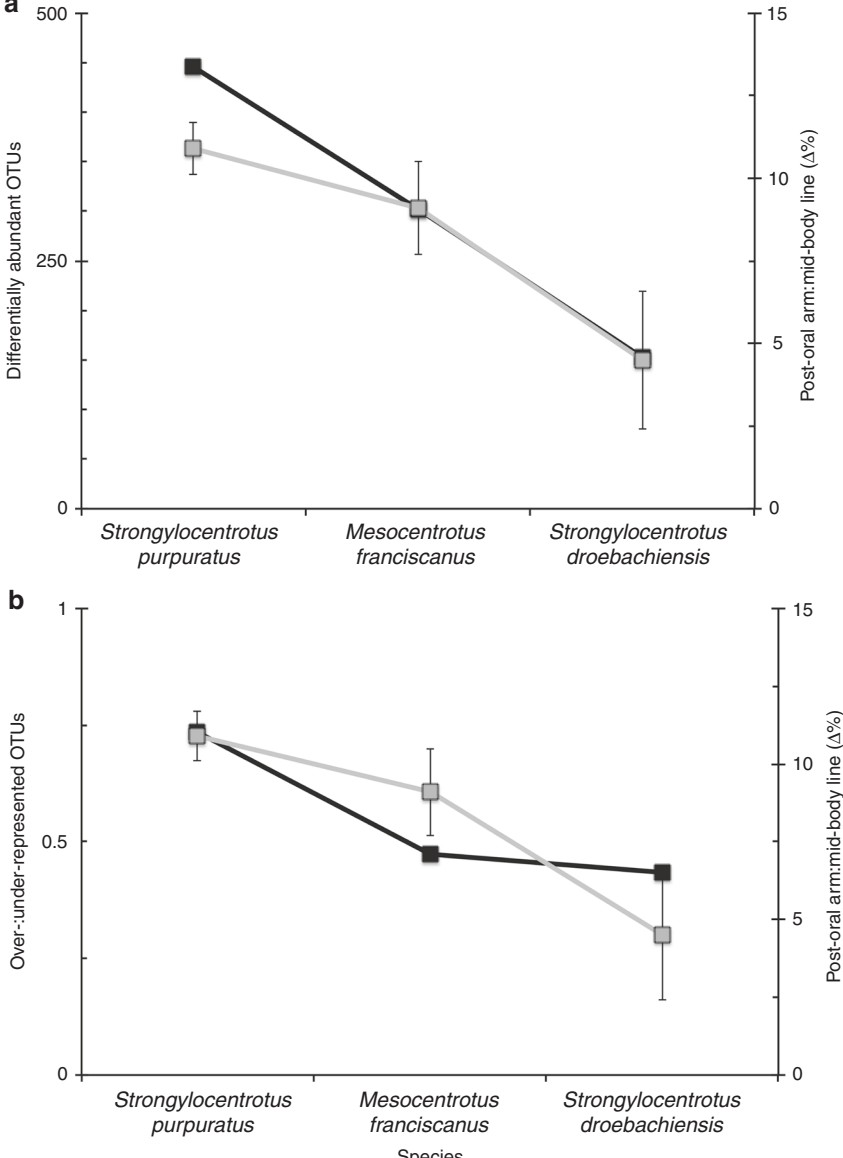

**Fig. 3** Differential abundance of OTUs along a morphological continuum for three species of echinoid larvae. Total (**a**) and ratio of (**b**) over- and under-represented OTUs associated with *Strongylocentrotus purpuratus*, *Mesocentrotus franciscanus*, and *S. droebachiensis* larvae following the expression of phenotypic plasticity (black) and in relation to the change in larval morphology (gray). Species on the *x* axis are organized from least to most maternal investment, a direct correlate of the expression of phenotypic plasticity

(i.e., PC1) distinct from phenotype (i.e., PC2) and time (i.e., PC3) (Fig. 5; ANOSIM, *S. purpuratus*: $p < 0.001$, *M. franciscanus*: $p < 0.001$), further supporting that echinoid larvae associate with phenotype-specific microbial communities.

Differences in the associated microbial communities of the echinoid larval host across plasticity states may also have been the result of differences in the microbial communities prior to feeding. To test this, we compared the host-associated microbiota of pre-feeding larvae, finding that each biological replicate varied slightly but were more similar to each other than to other species of pre-feeding larvae (i.e., species-specificity; $p < 0.004$; Supplementary Fig. 17–20). This result mirrors a phylosymbiotic pattern[32] (data not shown), although the number of echinoid species is insufficient for a robust comparison. Thus we observed no support that differences in microbial signatures across phenotypes, dietary states, and developmental stages were due to pretreatment differences.

Alternatively, these differences may have been the product of temporal shifts in the environmental microbiota during the course of the experiment. When comparing larval-associated and environmental microbiota from pre-feeding and late larval development, we observed that each species of pre-feeding (ANOSIM, $p < 0.001$) and post-feeding (ANOSIM, $p < 0.001$) larva was distinct from the environmental microbiota (Supplementary Fig. 21), suggesting that plasticity- and diet-specific microbial associates were unlikely to be the product of differential exposure to environmental microbiota. These results suggest that plasticity- and diet-specific microbial associates were unlikely to be the product of differential exposure to environmental microbiota.

**Bidirectional shifts in the associated microbial community.** Expression of nutritional plasticity can be reversible with a change in the feeding regime. To test whether the plasticity-

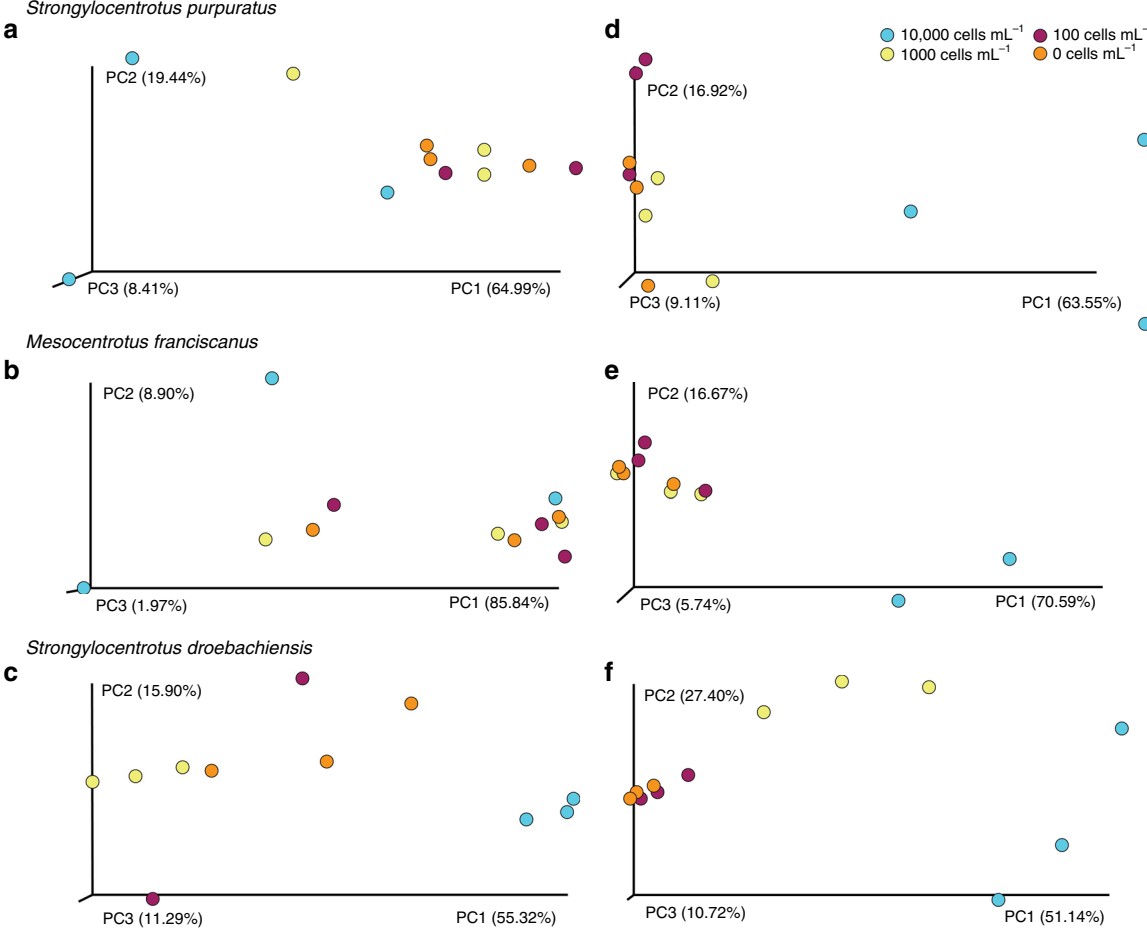

**Fig. 4** Induction of differential associations in the microbial community based on feeding environment for three species of echinoid larvae. Community similarity of the associated microbiota for *Strongylocentrotus purpuratus* (**a**, **d**), *Mesocentrotus franciscanus* (**b**, **e**), and *S. droebachiensis* (**c**, **f**) prior to (**a**–**c**) and post (**d**–**f**) expression of phenotypic plasticity, with larvae having been fed 10,000 (blue), 1000 (yellow), 100 (maroon), or 0 cells mL$^{-1}$ (orange)

associated microbial signature is reversible, *S. droebachiensis* larvae were starved (0 cells mL$^{-1}$) for 3 weeks then switched to an ad libitum diet (10,000 cells mL$^{-1}$) for 3 weeks (i.e., until metamorphosis).

The microbiome of larvae fed ad libitum followed a development-specific trajectory while starved larvae, as before, remained distinct from well-fed siblings and was similar to a starvation-specific microbial community (Figs. 4 and 6). The structure of the microbial community associated with early-stage larvae, as discussed above, were more similar to each other (weeks 1 and 2) than to late-stage larvae, independent of diet. Within the later larval stages, a division in community similarity was observed between starved larvae (weeks 3, 4, and 6) and well-fed larvae (Supplementary Fig. 22). Furthermore, when starved larvae were switched to a well-fed diet, their associated microbial communities became more similar to larvae fed ad libitum with time (Fig. 6; Supplementary Fig. 22), a trajectory congruent with developmental morphology (Fig. 1c).

## Discussion

Evolutionary and ecological theory predicts that variation in host-associated microbial communities corresponds with host phenotype[8,33,34]. If the impact of associated microbes was sufficient to contribute to fitness of the holobiont, we would predict that the host should be under selection to regulate what microbial species they associate in different environments[8,11,13,35]. Examples include, but are not limited to, aphids and *Buchnera*[36], the

bobtail squid *Euprymna* and *Vibrio fischeri*[37], and the parasitic wasp *Nasonia* and *Wolbachia*[38], and on a community level, the gut and root microbiome of many animals[15] and plants[39]. Previous studies, however, have not directly tested whether the microbiome correlates with environmentally induced morphological plasticity in adaptive characters. Morphological plasticity is present in many species and is likely adaptive by facilitating a better matching phenotype for increased performance in the environment experienced by the holobiont[1,4,8,9,34]. If the associated microbial community contributes to the relative fitness of the host experiencing a dynamic environment, we hypothesized that the community should shift with morphological plasticity.

Using larvae from three echinoid species ranging in their ability to express morphological plasticity, we observed that the microbiome predictably shifted for all three species of larvae. For each species, changes in the morphology of larvae experiencing food-restricted environments resulted in a corresponding shift in the microbial community. Interestingly, although similar patterns of differential association were observed between species of larvae, the microbial taxa were similar at higher taxonomic levels (e.g., phylum and class) but not at the OTU level. Moreover, for the species of larvae experiencing a coarse environmental shift (i.e., *S. droebachiensis* from unfed to high food), a phenotype-specific microbial signature was reversible, implying that a microbiome-based means of acclimating to environmental variation is bidirectional and, perhaps, a fluid component of hologenomic acclimation.

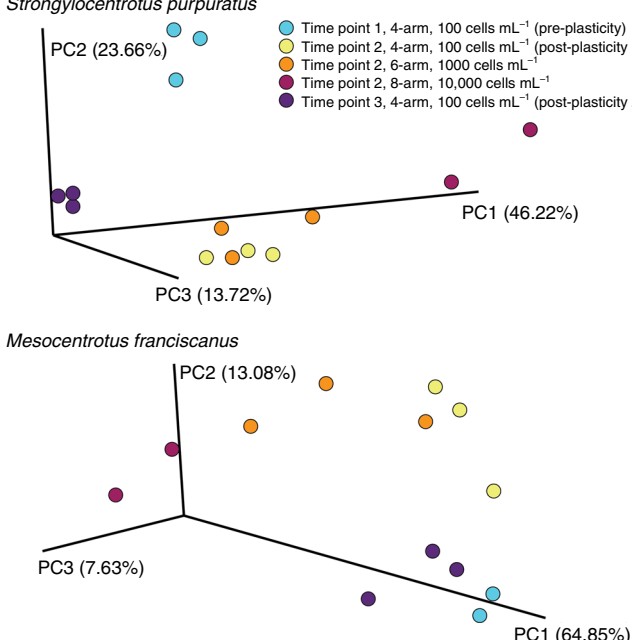

**Fig. 5** Decoupling phenotype-specific microbial communities from diet, development, and time for two species of echinoid larvae. Community similarity of the associated microbiota for *Strongylocentrotus purpuratus* (**a**) and *Mesocentrotus franciscanus* (**b**) larvae at the 8- (maroon), 6- (orange), and 4-arm (yellow) stage having been fed 10,000; 1000; and 100 cells mL$^{-1}$, respectively, in comparison with larvae pre-expression (blue) and post-expression (purple and yellow) of phenotypic plasticity

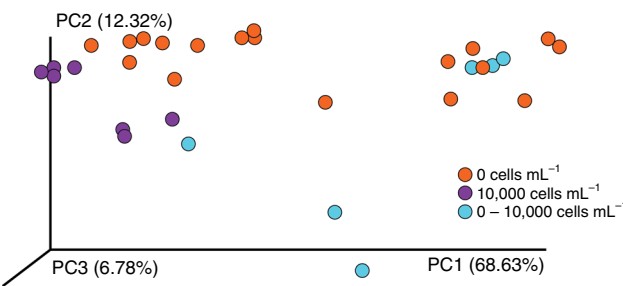

**Fig. 6** Bidirectional plasticity in the associated microbial communities for *Strongylocentrotus droebachiensis* larvae. Community similarity of the associated microbiota for *S. droebachiensis* larvae fed 10,000 cells mL$^{-1}$ (purple) until metamorphosis and 0 cells mL$^{-1}$ (orange) for 3 weeks, versus larvae fed 0 cells mL$^{-1}$ for 3 weeks, then switched to 10,000 cells mL$^{-1}$ until metamorphosis (blue)

Larvae of *S. purpuratus*, *M. franciscanus*, and *S. droebachiensis* partner with species-specific microbiota that exhibit similar patterns of differential association when exhibiting phenotypic plasticity. In light of species-specific patterning in larval-associated microbiota, we propose that the functional importance of the microbiome corresponding with phenotypic plasticity is similar between species while the bacterial taxa vary. Our hypothesis follows recent evidence that the microbial taxa associated with a host may not directly reflect the functional properties of that community[40]. Based on the degree to which larvae can exhibit morphological plasticity, we suggest that the particular functions of the microbial communities associated with *S. purpuratus* and *M. franciscanus* larvae expressing plasticity are more similar than that of *S. droebachiensis*[32].

Corresponding with a predicted convergence in functional properties of the host-associated microbial communities across phenotypes, shifts in these communities may be mediated by differential gene expression of the larval host. Previous transcriptomic comparisons have shown that *S. droebachiensis* larvae exhibit a broad transcriptomic response to differential feeding[23]. Following the expression of phenotypic plasticity, *S. droebachiensis* larvae downregulate genes associated with growth and metabolism while upregulating genes involved with neurogenesis and environmental sensing, immunity and defense, and longevity[23]. Interestingly, the predicted function of upregulated genes when larval echinoids undergo phenotypic plasticity also correspond with well-known functional properties of microbes[20,41–49]. Therefore, a functional-based approach[50] should be taken to determine whether the host gene expression or microbial interactions regulates, or perhaps directs, phenotypic plasticity.

The rate of environmental change and delay in the corresponding phenotypic response can limit the phenotype–environment match. Larvae of the sea urchin *Lytechinus variegatus* exposed to fine grain (2-day) variability in exogenous resources, for example, are unable to match phenotype with feeding regime because the delay required for phenotypic reconstruction exceeds the environmental variability[51]. However, differentially associating with microbial communities when experiencing similar fine grain environmental variation, as similarly shown here with *S. droebachiensis* (10,000 versus 0 cells mL$^{-1}$), may be quicker to modulate over short temporal oscillations than morphological changes, which are typically slow. Thus, when facing environments variability favoring the expression of alternate morphological traits, organisms may acclimate by differentially associating with microbial communities.

Phenotypic plasticity is common in animals and plants and, thus, our results of a phenotype-specific microbial community may be common when acclimating in variable environments. Polyphenism in anuran tadpoles, for example, is highly dependent on diet type, such that the morphology of carnivore and omnivore morphs from the same clutch differs considerably[52,53]. Namely, carnivorous tadpoles have a larger orbitohyoideus-to-snout-length ratio, enabling more efficient predation on their preferred dietary option[52,53]. Terrestrial plants, on the other hand, are highly plastic with regards to resource acquisition. For example, low-nutrient soil environments mediate increased growth in the roots and harvestable areas for the rhizosphere, whereas low levels of light results in an increase in leaf area[54]. Anuran tadpoles and plants, therefore, may serve as comparative systems for studying the hologenomic evolution of phenotypic plasticity and whether additional environmental cues select for shared and unique mechanisms associated with acclimation.

Taken together, the data presented here support the hypothesis that sea urchin larvae have a phenotype-specific microbial community and that morphological change is correlated with restructuring the associated microbial community. Future research should determine whether the bacterial associates and other type of microbes influence the expression of larval genes and what metabolites they contribute to the host will elucidate how these microbes may contribute to maximizing hologenomic fitness in a heterogeneous sea.

## Methods
**Adult urchin collection and larval rearing**. Adult urchins were collected from populations throughout the Salish Sea in April 2016. Specifically, individual *S. purpuratus* were hand-collected at low tide at Slip Point, Clallam Bay, WA (48° 15′39″ N, 124°15′03″ W) and transferred overnight to the Friday Harbor Laboratories (FHL; University of Washington; Friday Harbor, WA, USA). Similarly, *S. droebachiensis* were hand-collected at low tide, except at Cattle Point, San Juan Island, WA (48°27′00″ N, 122°57′43″ W), and were transferred to FHL within the hour. *M. franciscanus*, on the other hand, were collected by SCUBA off Bell Island, WA (48°35′49″ N, 122°58′55″ W) and transferred to FHL within 2 h. Collected

urchins were suspended in sub-tidal cages off the dock at FHL and fed *Nereocystis* spp. (sugar kelp) ad libitum until spawning 2 weeks later.

Adult urchins were spawned with a 1- to 2-mL intracoelomic injection of 0.50 M KCl. For each species, gametes from up to three males and three females were separately pooled. Fertilization of eggs and larval rearing followed Strathmann[55], except, to include the environmental microbiota, embryos and larvae were reared using 5.0-μm filtered seawater (FSW; instead of traditional filtration at 0.22-μm). Briefly, embryos were incubated in 1 L of FSW at ambient temperature and salinity (Supplementary Fig. 24) and, 2 h post-fertilization, were transferred to 3 L of FSW, divided into triplicates, and larval density was fixed to 2 larvae mL$^{-1}$, with subsequent dilutions with development. Larval cultures were given 90–95% water changes every other day.

Monocultures of *R. lens* were grown at room temperature with f/2 media and a combination of ambient and artificial lighting[56].

**Experimental feeding and larval morphometrics**. At 48 h post-fertilization, prism-stage larvae were divided into three replicate jars for each of the four experimental feeding treatments varying in *R. lens* quantity: 10,000; 1000; 100; or 0 cells mL$^{-1}$. For each species, larvae fed 10,000 cells mL$^{-1}$ were reared through metamorphosis while starved larvae were diet-restricted until developmental stasis was reached. Larvae ($n = 100$) of each species from all treatments and replicates were sampled weekly, preserved in RNAlater, and stored at $-20$ °C until extractions of nucleic acids were performed. We also tested for how diet shifts influence development and associated microbes in *S. droebachiensis* by starving larvae for 3 weeks and then switching them to 10,000 cells mL$^{-1}$ through metamorphosis. Larvae from this experiment were preserved, and the nucleic acids were extracted in an identical manner.

In addition to sampling larvae to assay the associated microbial communities, 20 larvae from a single replicate for each dietary treatment were sampled for morphometric analysis. Larvae were imaged using a compound microscope (Nikon Eclipse E600; camera: QImaging MicroPublisher 5.0 RTV) and morphometrics (length of larval body, post-oral arms, and stomach area; Supplementary Fig. 1D) were measured using ImageJ (NIH software, ver. 1.9.2)[57]. We tested whether larval morphology and stomach volume were influenced by differential feeding over time using a two-way ANOVA (JMP Pro v. 13). Where statistical differences were observed ($p < 0.05$), we used a post-hoc test to determine the affect at each time point and for each diet.

**Assaying microbial communities**. We extracted total DNA from larval samples using the GeneJet Genomic DNA Purification Kit (Thermo Scientific). For FSW samples, we extracted eDNA using the FastDNA Spin Kit for Soil (MP Biomedical). DNA was then quantified using the NanoDrop 2000 UV-Vis Spectrophotometer (Thermo Scientific) and diluted to 5 ng μL$^{-1}$ using RNase/DNase-free water.

Bacterial sequences were amplified using universal primers for the V3/V4 regions of the 16S rRNA gene (Forward: 5′CTACGGGNGGCWGCAG, Reverse: 5′ GACTACHVGGGTATCTAATCC)[58]. Products were purified using the Axygen AxyPrep Mag PCR Clean-up Kit (Axygen Scientific), indexed via PCR using the Nextera XT Index Kit V2 (Illumina Inc.), and then purified again. At each of these three clean-up states, fluorometric quantitation was performed using a Qubit (Life Technologies) and libraries were validated using a Bioanalyzer High Sensitivity DNA chip (Agilent Technologies). Illumina MiSeq sequencing was performed at the University of North Carolina at Charlotte.

Forward and reverse sequences were paired and trimmed using PEAR[59] and Trimmomatic[60], respectively, converted from fastq to fasta using custom script, and, prior to analysis of bacterial 16S rRNA sequences, chimeric sequences were detected using USEARCH[61] and removed using filter_fasta.py. Using QIIME 1.9.1[62], bacterial 16S rRNA sequences were analyzed and grouped into OTUs based on a minimum 97% similarity. The biom table generated by the pick_open_reference_otus.py script was filtered of OTUs with <10 reads as well as sequences matching chloroplast for cryptophytes (i.e., *R. lens*; Supplementary Data 1–3).

Using the filtered biom table and "biom summarize-table" function to count total sequences per sample, the rarefaction depth of 18,225 was determined and applied to all subsequent analyses (Supplementary Fig. 23). Beta diversity was calculated using the weighted UniFrac[63], and principal coordinate analyses were visualized in EMPeror[64] and stylized in Adobe Illustrator CS6. Community composition was generated using summarize_taxa_through_plots.py script and visualized using Prism 7 (GraphPad Software). Community similarity across phenotypes, dietary states, developmental stages, and their decoupling were compared statistically using an ANOSIM as part of the compare_categories.py script.

A step-by-step listing of QIIME scripts used to convert raw reads to OTUs for visualization of the data is available in Supplementary Note 1.

**Data availability**. The 16S rRNA data supporting the findings presented in this study are available on Dryad under the DOI, doi:10.5061/dryad.v7g08.

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

## Acknowledgements

We thank Jason Hodin and Billie Swalla for providing laboratory space; Morgan Eisenlord, Gustav Pauley, Katie Dobkowski, and Joe Gaydos for collecting urchins; Colette Feehan and Richard Strathmann for larval rearing advise; Daniel Janies for sequencing resources; Karen Lopez for technical assistance with sequencing; and Kevin Lambirth, Jason Macrander, and Andrew Brooks for bioinformatics assistance. T.J.C. was supported by an NSF Graduate Research Fellowship, Charles Lambert Memorial Endowment fellowship from the Friday Harbor Laboratories, and a Sigma Xi Grants-in-Aid of Research grant; A.M.R. was supported by NSF DEB1545539 and Human Frontier Science Program Award RGY0079/2016.

## Author contributions

T.J.C. conceived, designed, and performed the experiments and analyzed and interpreted the data. Both the authors discussed the results and wrote the manuscript.

## Additional information

**Competing interests:** The authors declare no competing interests.

