## [Peer Review File · Nature Communications]

Reviewers' comments:

Reviewer #1 (Remarks to the Author):

This is a well-written paper that articulates an important and overdue concept for testing: the contribution of microorganisms to phenotypic plasticity of the host. I was excited to read this paper as I expected to see evidence for how the microbial community contributed to increased fitness in a morphologically plastic invertebrate host. However, I was disappointed to find that this was in fact not tested and that the study was instead entirely correlative based on shifts in microbial community structure at the 16S rRNA gene level. While the amplicon analyses are robust and the Authors show good evidence for phenotype-specific microbial communities, there is little support for a cause:effect pathway influencing holobiont fitness. As there is no previous genomic reference or baseline functional microbiome analyses of these echinoderm species (well at least no papers were cited), there is no standing knowledge on the function of the microbiome for which to interpret these community shifts in an acclimatisation context.

Instead of taking a microbial survey approach and correlating this to the different morphological stages, more meaningful insights would have been derived if the Authors had undertaken a metatranscriptomic (or even metagenomic to infer the functional potential of the community), as was done for the host urchin in the publication by Carrier et al in 2015. The importance of undertaking a function-based approach to understanding holobiont acclimatisation was recently highlighted in an ISME J paper by Webster and Reusch, and the Authors should consider whether their sample set would enable them to experimentally test variation in host fitness due to altered microbial function. Another article by Theis and Bordenstein on the 10 principles of hologenomes describes how hologenomic plasticity must consider the genomic plasticity of the microbial associates, yet this was not tested here? With no knowledge of the level of functional equivalence in the microbiome of these echinoderm species, it is difficult to say anything about how the microbiome is actually related to acclimatisation (other than that the community correlates with developmental stage and diet which has been shown in numerous other studies with other model species ranging from hydra to humans).

I list a few additional considerations for the Authors below:

Please provide data from field samples of these developmental stages for comparison – without these it is not possible to assess the environmental relevance of the microbiome shifts.

It would be valuable to know the level of intra-species variability in the microbiome of these echinoderm species using replicate field samples from different sites / collection times etc.

For a 16S rRNA gene survey, the replication level is rather low.

The discussion over lines 238-258 is highly speculative and needs to be toned down as there is absolutely no evidence provided to show that the microbiome directly influenced host gene expression.

Reviewer #2 (Remarks to the Author):

This study focuses on the correlation and relationship between phenotypic plasticity and microbiome structure in the larvae of three species of sea urchins: *Strongylocentrotus purpuratus*, *Mesocentrotus franciscanus*, and *Strongylocentrotus droebachiensis*. The authors manipulated food availability over the span of 4 weeks in order to elicit different phenotypic traits in the larvae. They proceeded to

analyze the larvae's microbiomes across morphological states and diets to examine and determine the relationship between the microbial communities, host phenotype, and environmental factors (testing their hypothesis that holobionts follow a host genome-by microbial metagenome-by environment interaction in the context of phenotypic plasticity).

The questions posed in this study are original and interesting, and could go a long way in helping our understanding of the effect of environmental factors on the host-microbial networks. The methods used for testing are adequate and the authors do a good job in de-coupling each factor (phenotype, diet, development) when analyzing microbial responses.

However, authors should be careful with overstating the implications of their results; while their data do point to a clear relationship between morphological changes and shifts in the urchins' microbial communities, there is no clear evidence that indicates whether microbial shifts are causing the change in urchin phenotype. The authors state in their introduction that they aim to "test the hypothesis that microbial communities differentially associate with, and perhaps direct, the expression of phenotypic states," and in their discussion, they write that "data presented here support the hypothesis that the evolution of phenotypic plasticity is hologenomic." According to the authors, the hologenomic theory of phenotypic plasticity states that acclimation to environmental changes is the result of the microbial metagenome (GM) and is succeeded by the host genome (GH). In order to test for this kind of relationship, the authors would have to do similar experiments with much finer time scales/shorter sampling intervals in order to be able to detect whether the microbial shifts or the morphological changes come first, as well as doing a complete analysis on the functions of the microorganisms that are key players in these shifts to determine what effects, if any, they could have on phenotypic characteristics of the urchin larvae.

I think the manuscript is still worthwhile of publication in Nature Communications as long as some of the potential caveats of the study are discussed.

General comments:

Figures can be difficult to understand, such as Figure 3, and should be explained further to make sure readers understand the data being presented. Most figures would also benefit from having a figure legend with a description of what each color is meant to represent, as well as including the species names of each urchin.

Authors often write long, jargon-y sentences to explain their experiment and ideas, on occasion using words incorrectly. These are difficult to understand and confusing, thereby making the manuscript and the authors' ideas less accessible. Authors should consider going over the manuscript and these sentences in particular to make sure they are getting their point across in the clearest and most readable way possible.

Examples:

Lines 48 to 51 contain a run-on sentence that is difficult to understand.

The words "discern" and "broader inference" on lines 63 and 66 are used incorrectly.

On line 72 the authors refer to the microbiome as a "genomic component" and it is unclear what they mean by this; is the implication that microorganisms act the way a genomic component would, if you were to think of the holobiont as one big genome? Or are they implying that the microbial genomes are changing as well?

Reviewer #3 (Remarks to the Author):

In this manuscript, the authors assessed the microbiome composition associated to larvae of three sea urchin species, at distinct stages of phenotypes as induced by varied abundance of cryptophyte cells of *Rhodomonas lens* (as feed). The results indicate that microbiome compositions may contribute to phenotypic plasticity in these larvae, dominated by alpha- and gamma-proteobacteria.

These findings are novel, and potentially of broad interest to the research community in marine biology and ecology, and microbial evolution. However, the presentation and interpretation of the results appear problematic.

Much of the results (and many p-values) are presented with little or no context. Many figures do not contain legend for the different colours/data points, and labels, making interpretation difficult. The experimental design should be briefly described before the results are presented. Some interpretations appear unreasonable, particularly the trends observed based on the three species - line graphs are inappropriate (see my comments re: Figure 3 below). I believe the results of the microbiome compositions are more relevant and important in this study, thus they should be presented as main figures. Some of the PCA plots in the main figures (Figs 2, 4, 5, 6) are perhaps better presented as supplemental figures.

In addition, much of the discussion surrounds the potential gene functions in the sea urchins (not done in this study) that may contribute to variations in the microbiome compositions, and it does not specifically address microbiome variation. In addition, to relate these results to hologenomes (L305-306), or to claim to have shown hologenomic plasticity (L21) is rather far-fetched without any genome data/analysis from the sea urchins. I suggest the authors to tone down their claims (see below). A hologenome should consist of all components of the holobiont (i.e. the sea urchin and the associated microbiome).

I list my detailed comments below, which I believe would improve the manuscript. The line numbers follow the generated PDF:

1. L14-16: This statement seems too strong. I believe the study of phenotypic variations associated with microbiota have been studied to some extent in algae. See Brodie et al. 2017 in *New Phytologist* (doi:10.1111/nph.14760).
2. L20 and throughout the text: the symbol of "upsilon" is misused as "gamma" to describe gamma-proteobacteria. I am not aware of any epsilon-proteobacteria. Perhaps it would be better to spell out "alpha" and "gamma" in the Abstract.
3. L21: you have not shown hologenomic plasticity in this work.
4. L22-L25: I believe that "your results" may support the hypothesis, not "you" per se. The sentence also appears problematic, because simply by "manipulating diet quantity over time" would not have supported or revealed anything - the observations/results from the experiment did.
5. L28: this is the first time I encounter "GxE" defined as "genome-by-environment". It's commonly defined as "gene-by-environment"; this may create confusion in the literature. Given that you only used this term twice in the main text, there may not be a need to invoke the term "GxE".

6. L32: "natural selection"?
7. L33: "Much of ecological and evolutionary theory"?
8. L36: I note that some holobionts include other eukaryotes as well, like lichen, corals etc.
9. L50-51: "feeding (planktotrophic) marine invertebrate larvae" - did you mean "feeding (planktotrophic) larvae of marine invertebrates" or "larvae of feeding (planktotrophic) marine invertebrate"? I imagine all invertebrates feed?
10. Results (general): it remains unclear throughout the study until L340-341 that the larvae were fed with cells of *Rhodomonas lens*. This, and the brief experimental design should be clarified in the beginning of the Results section, so the readers can understand the presented results better (per my concern above).
11. Supp Table 1: what is "--" in the table? No arms observed? Did they die?
12. Supp Tables 2, 3: F-ratio, SS are not defined. What are these?
13. L83: Supp Fig 2 shows stomach volume of the larvae, which is not mentioned in the text, making the citation here irrelevant.
14. L85-86: How can Figure 1 (that shows the post-oral arm:mid-body line ratios), and Supp Fig 3 (that shows egg diameter of unfertilised zygotes) explain the inverse correlation of morphological change to the degree of "maternal investment"? It is not until L221 where you relate maternal investment to egg size. Besides, three measurements in Supp Fig 3 could perhaps be better represented as a table.
15. L88-89: the "morphological plasticity" for these larvae (i.e. those fed with 100 cells/mL) is not obvious in Supp Figure 2A. The ratios seem pretty similar to me.
16. L91-92: the increase between weeks 1 versus 2, or between weeks 1 versus 3, is not obvious in Supp Figure 2B.
17. L95-96: the ratio of these larvae (in Fig 1C and Supp Fig 2C) appears to show signs of decreasing instead of increasing as the text indicates.
18. L100-104: this sentence is convoluted.
19. L81-83: not all larvae show higher post-oral arm:mid-body line ratio with time.
20. Figures (general): it would be very helpful if the panels were labelled, e.g. with each urchin species, and a colour legend is presented for the bar charts/line graphs on the figures. I find it tricky to go through the main text, the figure, and the figure legend (in three different spots) to try to interpret each graph.
21. Figure 3: Line chart here is not appropriate (this also applies to Supp Figs S7A-S7C). A line suggests that there is a trend of these values going from one echinoid species to another, but the presentation order of these three species is arbitrary (i.e. changing the order of these three species will change the trend one observed on the line charts). In addition, OTUs would not have been up- or down-regulated (as indicated in the figure legend and text in L111) - they are not the same as gene

expression. I believe you meant over- or under-represented OTUs - but it bears the question of over-/under- relative to what exactly? This needs to be clarified in the text.

There are three measurements shown in the two graphs in Figure 3 (the post-oral arm:mid-body line ratio is repeated in each graph) - none of these measurements is described/mentioned in the figure legend. The composite graphs are confusing as it is unclear from the legend which data points/line represent which measurement on the graph (i.e. which corresponds to the y-axis on the left, and to the y-axis on the right). Legend for the data points would be helpful (see my comments re: all figures above). Also, there is a difference between a ratio (no upper limit) and percentage (ranges 0-100%).

22. Supp Figures 7B and 7C: I believe the red bars in 7C depict over-represented OTUs, and the blue bars represent the under-represented OTUs, and that these correspond to the Over:Under ratios shown in 7B? This is unclear from the figure legend/text.

23. L102-104: Figure 2 with six panels is simply mentioned here once. Figures 2D, 2E and 2F were mentioned later in L119. Should this be a main figure or better presented as a Supplemental Figure? Also, in Figures 2D-2F, what do A and B above the bars denote?

24. L105-113: The trends described here assume the presentation order of the three species is meaningful (e.g. a time series, or progression of development stages), which is not the case. The text here is misleading and the results are interpreted inadequately. See my comments above about Figure 3.

25. L116-128: it is unclear why the different bacterial phyla were chosen in Supp Figs 8A-8C for each echinoid species. They represent different resolutions: some are at family level (Bradyrhizobiaceae, Colwelliaceae), some are at genus level (Oleispira, Pseudomonas). Are these the most represented bacterial taxa in each species?

26. Supp Figs 10, 12 and 14 show distributions of bacterial taxa in each of the three echinoid species before expression of phenotypic plasticity. Those after expression of phenotypic plasticity are shown in Supp Figs 11, 13, 15. It would be of tremendous help if the colour scheme for the different bacterial phyla were consistent across these graphs, especially if we were to assess the before-and-after comparison for a single sea urchin species. The before-and-after snapshot is more interesting than the various PCA plots - perhaps Supp Figs 10 and 11 could be presented as a main figure instead.

27. L175: Supp Figs 18-21: similar to my comments above (Supp Figures 10-15) regarding the consistency of colour schemes. The presentation of results appears to have skipped Supp Fig 17, which is presented later in L201-202.

28. L176: "Supplemental Figure 22". Also, in the figure legend, did you mean "COI sequences"? "CO1" is not a standard gene name. In addition, a phylogenetic relationship based on only three sequences/taxa is not meaningful. How was the tree reconstructed? This should be at least mentioned in the text if not in the Methods section.

29. L232-280: you reviewed the literature the potential gene functions of sea urchins that may play a role in morphological plasticity in diet-restricted environments, and how this may contribute to changes in the associated microbiomes. It remains unclear how this is relevant to this study, as you did not assess the gene expression of sea urchins under the conditions in your experiment.

30. L305-306: the results suggest that microbiome compositions may contribute to phenotypic plasticity in the echinoid larvae, but they do not "support the hypothesis that the evolution of

phenotypic plasticity in hologenomic" as the text currently claims. Also, this stated hypothesis remains vague. To really test the hypothesis related to hologenomes, hologenome data (i.e. genes, functions inferred from the genomes of the sea urchins and the microbiomes) are necessary.

31. L327-329: this sentence appears incomplete.

32. L333-334: it is unclear what you mean by "90-95% water changes" every other day. Did you replenish/replace 90-95% of the seawater every other day?

33. L368, L377-378: the S in "16S" is always in capital.

34. L373-374: Knowing the length of these MiSeq sequence reads would be helpful. I imagine there is no assembly step involved in this work?

Thanks,
Cheong Xin Chan

NCOMMS-17-23501-T

Reviewers' comments:

Reviewer #1 (Remarks to the Author):

This is a well-written paper that articulates an important and overdue concept for testing: the contribution of microorganisms to phenotypic plasticity of the host. I was excited to read this paper as I expected to see evidence for how the microbial community contributed to increased fitness in a morphologically plastic invertebrate host. However, I was disappointed to find that this was in fact not tested and that the study was instead entirely correlative based on shifts in microbial community structure at the 16S rRNA gene level. While the amplicon analyses are robust and the Authors show good evidence for phenotype-specific microbial communities, there is little support for a cause:effect pathway influencing holobiont fitness. As there is no previous genomic reference or baseline functional microbiome analyses of these echinoderm species (well at least no papers were cited), there is no standing knowledge on the function of the microbiome for which to interpret these community shifts in an acclimatisation context.

We thank Reviewer 1 for their enthusiasm for our study, which is the first test of shifts in the host-associated microbial community across host phenotypic states. The Reviewer is correct that we have no baseline from previous publications to compare our results and, furthermore, there are certainly no data on potential functions of these associated bacteria to assess their contribution to the fitness of the larval holobiont.

The significance in the present study, however, is the outcome for a convergence in phenotype-specific microbial community across multiple invertebrate hosts, and that this is bi-directional when the environment is variable. While we did not study functional properties of the associated bacterial taxa, a similar community-level shifts in microbiota across urchin species does suggest a common response that future research would profit from a functional determination.

Instead of taking a microbial survey approach and correlating this to the different morphological stages, more meaningful insights would have been derived if the Authors had undertaken a metatranscriptomic (or even metagenomic to infer the functional potential of the community), as was done for the host urchin in the publication by Carrier et al in 2015. The importance of undertaking a function-based approach to understanding holobiont acclimatisation was recently highlighted in an ISME J paper by Webster and Reusch, and the Authors should consider whether their sample set would enable them to experimentally test variation in host fitness due to altered microbial function. Another article by Theis and Bordenstein on the 10 principles of hologenomes describes how hologenomic plasticity must consider the genomic plasticity of the microbial associates, yet this was not tested here? With no knowledge of the level of functional equivalence in the microbiome of these echinoderm species, it is difficult to say anything about how the microbiome is actually related to acclimatisation (other than that the community correlates with developmental stage and diet which has been shown in numerous other studies with other model species ranging from hydra to humans).

We agree that the present study may be appropriately complemented with meta-genomics and/or meta-transcriptomics to connect phenotype-specific microbial communities with the functional properties of the associated microbial community. Our results do indicate particular time points-diet combinations that would be insightful to conduct these assays.

As described in our "General Response Overview," a major limitation to using marine invertebrate larvae for -omic-centric studies is collecting sufficient tissue, especially in a field setting, as done here. For these experiments, we invested our sampling effort to study the maximum time points and diets to characterize potential phenotype-specific shifts independently of develop, diet, and ecological drift. This, in turn, substituted the potential to collect samples for meta-genomics or meta-transcriptomics. Moreover, independent of this limitation, both *Strongylocentrotus droebachiensis* and *Mesocentrotus franciscanus* lack reference transcriptome databases in which to discern whether meta-genomic and/or meta-transcriptomic data was of animal, bacterial, or foreign-origin. Maturation of these databases would result in a more functional comparison geared towards holobiont fitness.

Lastly, we are familiar with both papers the reviewer refers to and agree with their arguments for a function-based and hologenomic approaches to understanding holobiont acclimatization. Again, as stated previously, successful application of these approaches requires sufficient tissue from which to extract the nucleic acids and rich reference databases to discern function; both are inadequate at this time.

I list a few additional considerations for the Authors below:

Please provide data from field samples of these developmental stages for comparison – without these it is not possible to assess the environmental relevance of the microbiome shifts.

Historically, to study early-stage marine invertebrates, larvae have been reared in laboratory jars in a microbe-restricted (*i.e.*, 0.22- μ m filtered seawater) environment. Using 5- μ m filtered seawater, we deviated from this protocol to include the environmental microbiota that would best replicate a field setting. The collection of early-stage marine invertebrates from the coastal or open ocean, whether that is an immature embryo or competent larva, is a near insurmountable feat. This is primarily due to the fact that most larvae are less than 1-mm in length.

Collecting field samples, therefore, requires exceptional effort through plankton sampling in the open ocean, which in the most successful case would only result in a few larvae with unknown species identity. Thus, field collected stages for each species reported in our study are impractical and potentially impossible. Our effort to maintain some of the environmental-relevance by using coarse filtered natural seawater is a logistically feasible compromise to maintain most of the natural environmental microbiota while also having the number of individuals required our sampling design. Moreover, as noted in our "General Response Overview," this system is restricted by rearing capacity, not enabling for a time series to be collected alongside a plasticity experiment.

It would be valuable to know the level of intra-species variability in the microbiome of

these echinoderm species using replicate field samples from different sites / collection times etc.

We agree with the Reviewer that this would indeed be an interesting insight. These species of sea urchin, however, reproduce annually during a short window from early- to mid-spring and experiments—entail near around-the-clock husbandry—to measure these morphological responses over a month. Thus, experimentation is restricted to a maximum to one site per year. In order to compare multiple sites or collection times for multiple species, we would need to replicate this experiment over at least four future years, which we feel is beyond the scope of what can be reasonably done.

For a 16S rRNA gene survey, the replication level is rather low.

Aside from the logistical limitations to the number of larvae that can be reared per culturing jar—as discussed above—the number of culturing jars per field season is heavily restricted, as well. Provided that we have three species and four diets, we were restricted to three biological replicates, totaling 36 jars and ~36,000 larvae. To increase the replication level either number of species or diets would be reduced. We would likely have been unable to detect a phenotype-specific microbial community and, subsequently, reinforced with multiple additional species.

Given this, we agree with the Reviewer that an increased biological replication level would instill more confidence in the validity of our data. We wish to express the value in that a phenotype-specific microbial community was supported in three species of sea urchin larva. Moreover, although most comparisons were supported in triplicate, each biological replication was a population of 100 larvae.

The discussion over lines 238-258 is highly speculative and needs to be toned down as there is absolutely no evidence provided to show that the microbiome directly influenced host gene expression.

We have modified the Discussion as the reviewer suggested and now feel it better represents the scope of the project and highlights the shifts in the microbial community in place of speculation about host gene expression. We have maintained some discussion of host gene expression to provide a potential mechanism that we, or others, could pursue in future experiments.

Reviewer #2 (Remarks to the Author):

This study focuses on the correlation and relationship between phenotypic plasticity and microbiome structure in the larvae of three species of sea urchins: *Strongylocentrotus purpuratus*, *Mesocentrotus franciscanus*, and *Strongylocentrotus droebachiensis*. The authors manipulated food availability over the span of 4 weeks in order to elicit different phenotypic traits in the larvae. They proceeded to analyze the larvae's microbiomes across morphological states and diets to examine and determine the relationship between the microbial communities, host phenotype, and environmental factors (testing their hypothesis that holobionts follow a host genome-by microbial metagenome-by environment interaction in the context of phenotypic plasticity).

The questions posed in this study are original and interesting, and could go a long way in helping our understanding of the effect of environmental factors on the host-microbial networks. The methods used for testing are adequate and the authors do a good job in decoupling each factor (phenotype, diet, development) when analyzing microbial responses.

We thank the reviewer for their assessment that our study makes a unique contribution to our understanding of host-microbe interactions in the environment.

However, authors should be careful with overstating the implications of their results; while their data do point to a clear relationship between morphological changes and shifts in the urchins' microbial communities, there is no clear evidence that indicates whether microbial shifts are causing the change in urchin phenotype. The authors state in their introduction that they aim to “test the hypothesis that microbial communities differentially associate with, and perhaps direct, the expression of phenotypic states,” and in their discussion, they write that “data presented here support the hypothesis that the evolution of phenotypic plasticity is hologenomic.” According to the authors, the hologenomic theory of phenotypic plasticity states that acclimation to environmental changes is the result of the microbial metagenome (GM) and is succeeded by the host genome (GH). In order to test for this kind of relationship, the authors would have to do similar experiments with much finer time scales/shorter sampling intervals in order to be able to detect whether the microbial shifts or the morphological changes come first, as well as doing a complete analysis on the functions of the microorganisms that are key players in these shifts to determine what effects, if any, they could have on phenotypic characteristics of the urchin larvae.

We thank the reviewer for these comments and we agree. We have modified the Discussion considerably to more carefully restrict this discussion to the data highlighted in the study. We have provided responses to the number of samples and our ability to measure functional variation (e.g., metatranscriptomics) in response to Reviewer 1 and in our Overview section above.

I think the manuscript is still worthwhile of publication in Nature Communications as long as some of the potential caveats of the study are discussed.

General comments:

Figures can be difficult to understand, such as Figure 3, and should be explained further to make sure readers understand the data being presented. Most figures would also benefit from having a figure legend with a description of what each color is meant to represent, as well as including the species names of each urchin.

We agree with the Reviewer and have made these suggested changes to the figure legends.

Authors often write long, jargon-y sentences to explain their experiment and ideas, on occasion using words incorrectly. These are difficult to understand and confusing, thereby making the manuscript and the authors' ideas less accessible. Authors should consider going over the manuscript and these sentences in particular to make sure they are getting their point across in the clearest and most readable way possible.

Examples:

Lines 48 to 51 contain a run-on sentence that is difficult to understand.

The words “discern” and “broader inference” on lines 63 and 66 are used incorrectly.

On line 72 the authors refer to the microbiome as a “genomic component” and it is unclear what they mean by this; is the implication that microorganisms act the way a genomic component would, if you were to think of the holobiont as one big genome? Or are they implying that the microbial genomes are changing as well?

We have gone through the manuscript and revised the writing to make sentences shorter and more direct. For the last example by Reviewer 2, we are referring to the combined genomes of all associated microorganisms when we use the term "genomic component." This usage is consistent with the hologenome concept originally defined by Zilbert-Rosenberg and Rosenberg (2008), Theis and Bordenstein (2016), and others.

Reviewer #3 (Remarks to the Author):

In this manuscript, the authors assessed the microbiome composition associated to larvae of three sea urchin species, at distinct stages of phenotypes as induced by varied abundance of cryptophyte cells of *Rhodomonas lens* (as feed). The results indicate that microbiome compositions may contribute to phenotypic plasticity in these larvae, dominated by alpha- and gamma-proteobacteria.

These findings are novel, and potentially of broad interest to the research community in marine biology and ecology, and microbial evolution. However, the presentation and interpretation of the results appear problematic.

We thank the Reviewer for their assessment that our study is novel and of broad interest in various disciplines. We address their listed concerns with the presentation and data interpretation in the detailed comments section below.

Much of the results (and many p-values) are presented with little or no context. Many figures do not contain legend for the different colours/data points, and labels, making interpretation difficult. The experimental design should be briefly described before the results are presented. Some interpretations appear unreasonable, particularly the trends observed based on the three species - line graphs are inappropriate (see my comments re: Figure 3 below). I believe the results of the microbiome compositions are more relevant and important in this study, thus they should be presented as main figures. Some of the PCA plots in the main figures (Figs 2, 4, 5, 6) are perhaps better presented as supplemental figures.

We provided responses to many of these comments below. We thank the reviewer for these suggestions and have incorporated many of them into the revised manuscript.

In addition, much of the discussion surrounds the potential gene functions in the sea urchins (not done in this study) that may contribute to variations in the microbiome compositions, and it does not specifically address microbiome variation. In addition, to relate these results to hologenomes (L305-306), or to claim to have shown hologenomic plasticity (L21) is rather far-fetched without any genome data/analysis from the sea urchins. I suggest the authors to tone down their claims (see below). A hologenome should consist of all components of the holobiont (i.e. the sea urchin and the associated microbiome).

Similar comments and suggestions were provided by both Reviewer 1 and 2, and, in response, we have modified the Discussion to reduce dialog on host gene expression and related hypotheses regarding the hologenome because we did not measure them directly.

I list my detailed comments below, which I believe would improve the manuscript. The line numbers follow the generated PDF:

1. L14-16: This statement seems too strong. I believe the study of phenotypic variations

associated with microbiota have been studied to some extent in algae. See Brodie et al. 2017 in *New Phytologist* (doi:10.1111/nph.14760).

Brodie et al. (2017) in *New Phytologist* provides an extensive review on algal origins and evolution. In this review, the authors do not mention plasticity and only mention phenotype once, and in that single case they refer to “life-history phenotypes.” In the context of the sea urchin larval system, and similarly for diverse taxa, life-history phenotypes are landmark phenotypes over the course of development, such as the transitions from larva to juvenile. Phenotypic plasticity, on the other hand, is a biologically related but a distinct subject for which a single life-history stage (e.g., larva) expresses multiple phenotypes. Therefore, due to this distinction, we have decided to leave this statement as is.

2. L20 and throughout the text: the symbol of “upsilon” is misused as “gamma” to describe gamma-proteobacteria. I am not aware of any epsilon-proteobacteria. Perhaps it would be better to spell out “alpha” and “gamma” in the Abstract.

We apologize for this mistake and thank the reviewer for catching it. We have changed all “Υ” to “γ.”

3. L21: you have not shown hologenomic plasticity in this work.

We agree with Reviewer 3 and have modified this sentence in the revised manuscript.

4. L22-L25: I believe that “your results” may support the hypothesis, not “you” per se. The sentence also appears problematic, because simply by “manipulating diet quantity over time” would not have supported or revealed anything - the observations/results from the experiment did.

As stated above: We agree with Reviewer 3 and have modified this sentence in the revised manuscript.

5. L28: this is the first time I encounter “GxE” defined as “genome-by-environment”. It’s commonly defined as “gene-by-environment”; this may create confusion in the literature. Given that you only used this term twice in the main text, there may not be a need to invoke the term “GxE”.

We agree with the Reviewer and have replaced ‘GxE’ with ‘genotype-by-environment’ to avoid confusion.

6. L32: “natural selection”?

As reviewed in citation ‘4’, and covered by other reviews (e.g., Agrawal, 2001, DOI: 10.1126/science.1060701), many instances of phenotypic plasticity are adaptive and empirical work supports that this trait can be selected for (e.g., Buskirk and Relyea, 1998, doi.org/10.1111/j.1095-8312.1998.tb01144.x). This supporting literature is summarized in L29-32.

7. L33: “Much of ecological and evolutionary theory”?

We agree that this phrase was vague and did not clearly summarize our intent for this sentence. **Therefore, we have modified this sentence in the revised manuscript.**

8. L36: I note that some holobionts include other eukaryotes as well, like lichen, corals etc.

We agree that holobionts could include all organisms in symbiosis even if our study is focused on animals and bacteria. We have modified the first sentence of the second paragraph in the revised manuscript.

9. L50-51: “feeding (planktotrophic) marine invertebrate larvae” - did you mean “feeding (planktotrophic) larvae of marine invertebrates” or “larvae of feeding (planktotrophic) marine invertebrate”? I imagine all invertebrates feed?

We agree that this phrase was vague and did not clearly summarize our intent for this sentence. To reflect this, we have changed “feeding (planktotrophic) marine invertebrate larvae” to “feeding (planktotrophic) larvae of marine invertebrate,” as suggested.

10. Results (general): it remains unclear throughout the study until L340-341 that the larvae were fed with cells of *Rhodomonas lens*. This, and the brief experimental design should be clarified in the beginning of the Results section, so the readers can understand the presented results better (per my concern above).

We agree with Reviewer 3, and have modified the first sentence of section *Larval morphometric* in the revised manuscript.

11. Supp Table 1: what is “--” in the table? No arms observed? Did they die?

As part of the Supplemental Table 1 heading, we have added “(‘--’ designating no sample).” Samples were not collected at these points because larvae had undergone metamorphosis (e.g.: *S. droebachiensis*, diet = 10,000, weeks 3-5) or larvae had died due to starvation (e.g.: *S. purpuratus*, diet = 0, week 4).

12. Supp Tables 2, 3: F-ratio, SS are not defined. What are these?

F-ratio (f-statistic) and SS (sum-of-squares) are statistical values as part of an ANOVA table and are commonly presented alongside p-values and *df*. We have defined these values as a footnote for each Supplemental ANOVA table.

13. L83: Supp Fig 2 shows stomach volume of the larvae, which is not mentioned in the text, making the citation here irrelevant.

We measured stomach size data as part of our morphometric analyses. Miner 2005 (doi.org/10.1016/j.jembe.2004.09.011)—published a paper on the trade-off between arm

length (*i.e.*, the expression of phenotypic plasticity) and stomach volume, which has been observed broadly in echinoderms. Our data are consistent with this result. Because this result is largely confirmatory of prior publications, we decided to present as supplemental data to keep the focus of the manuscript on the more original contributions (microbes).

To acknowledge this trade off, we have modified L58-61 in the Introduction of the revised manuscript. Moreover, of the six citations for this sentence, references 22, 24, and 25 use stomach dimensions in the quantification for plasticity state.

14. L85-86: How can Figure 1 (that shows the post-oral arm:mid-body line ratios), and Supp Fig 3 (that shows egg diameter of unfertilised zygotes) explain the inverse correlation of morphological change to the degree of “maternal investment”? It is not until L221 where you relate maternal investment to egg size. Besides, three measurements in Supp Fig 3 could perhaps be better represented as a table.

There is a well-established relationship between egg size (the primary proxy for maternal/energetic investment) and the expression of phenotypic plasticity (*e.g.*, McAlister 2007, doi.org/10.1016/j.jembe.2007.08.009). We agree that the data in Supplemental Figure 3 could be a table, but we prefer to present as a figure because it shows clear differences between the three sea urchin species.

15. L88-89: the “morphological plasticity” for these larvae (*i.e.* those fed with 100 cells/mL) is not obvious in Supp Figure 2A. The ratios seem pretty similar to me.

Our response to inquiry “13” addresses this inquiry, but we wish to add one additional detail on the biology of phenotypic plasticity in echinoderm larval. Along the phenotypic continuum, post-oral arm and mid-body lengths are largely considered continuous. Stomach volume, on the other hand, is primarily dependent on feeding state, such that well-fed larvae have a large stomach and larvae under low food conditions have highly reduced stomach. This pattern is seen throughout the literature and was consistent here, and thus is congruent with the expression of phenotypic plasticity in echinoderm larvae.

16. L91-92: the increase between weeks 1 versus 2, or between weeks 1 versus 3, is not obvious in Supp Figure 2B.

As clarified in inquiry “13,” stomach volume is largely similar across the lower feeding treatments (here, 1,000, 100, and 0 cells•mL⁻¹) as well as with time (see Supplemental Figure 2B in comparing weeks 1 versus 2 and weeks 1 versus 3). Moreover, the morphological change refers to the increase in post-oral arm length relative to the larval body (*i.e.*, increasing the relative feeding apparatus). For *M. franciscanus* (Figure 1B; Supplemental Figure 2B), the trade-off between stomach volume and arm-length, as described above, holds.

Both ‘Figure 1B’ and ‘Supplemental Figure 2B’ were mentioned here, as compared to only ‘Figure 1B,’ because the expression of morphological plasticity refers to lengthening of the post-oral arms (*i.e.*, Figure 1B) and the substantial reduction in stomach volume (*i.e.*, Supplemental Figure 2B). Thus, for all references in the text pertaining to

phenotypic plasticity of these larvae, Figure 1 and Supplemental Figure 2 referenced together.

17. L95-96: the ratio of these larvae (in Fig 1C and Supp Fig 2C) appears to show signs of decreasing instead of increasing as the text indicates.

As detailed in the manuscript and in our responses above, many species of sea urchin larvae increase the size their feeding apparatus (*i.e.*, express phenotypic plasticity) in the face of starvation. If starvation is maintained for a few weeks, then these larvae enter a developmental stasis to prolong life in the plankton (Carrier *et al.*, 2015). If larvae do not encounter food before energetic reserves are depleted larvae perish. One of these energetic reserves is the post-oral arms; thus, on the onset of persistent nutritional stress, sea urchin larvae (here, *S. droebachiensis*) may absorb the post-oral arms. We observed this specifically in Figure 1C but, nevertheless, *S. droebachiensis* expressed phenotypic plasticity when fed 1,000 cells•mL⁻¹ following two versus three or four weeks of diet-restriction.

18. L100-104: this sentence is convoluted.

We agree with Reviewer 3, and have modified this sentence in the revised manuscript.

19. L81-83: not all larvae show higher post-oral arm:mid-body line ratio with time.

The expression of phenotypic plasticity (*i.e.*, higher post-oral arm:mid-body line ratio), as mentioned above, is dependent on the feeding regime as well as on time.

It is well-established that larvae fed *ad libitum* have reduce their post-oral arm:mid-body line ratio with time due to the majority of exogenous resource being allocated towards the larval body. Larvae cultured on a restricted diet, on the other hand, grow in overall size but have relatively long feeding arms. This ratio, however, will decrease later in developmental time when larvae are starved for several weeks because the arm tissues are resorbed, as larvae utilize remaining energetic reserves. These larvae later die due to lack of exogenous energy.

Larvae for each species of sea urchin fed 10,000 and, in some cases, 1,000 cells•mL⁻¹ would be expected to show a lower post-oral arm:mid-body line ratio with time because exogenous food is sufficient for maximizing developmental rate.

20. Figures (general): it would be very helpful if the panels were labelled, e.g. with each urchin species, and a colour legend is presented for the barcharts/line graphs on the figures. I find it tricky to go through the main text, the figure, and the figure legend (in three different spots) to try to interpret each graph.

We agree with the reviewer that additional labels would benefit the impact of the figures. In all cases, legends and/or labels were added to the figures. Please also see related response to Reviewer 1 above.

21. Figure 3: Line chart here is be appropriate (this also applies to Supp Figs S7A-S7C). A

line suggests that there is a trend of these values going from one echinoid species to another, but the presentation order of these three species is arbitrary (i.e. changing the order of these three species will change the trend one observed on the line charts). In addition, OTUs would not have been up- or down-regulated (as indicated in the figure legend and text in L111) - they are not the same as gene expression. I believe you meant over- or under-represented OTUs - but it bears the question of over-/under- relative to what exactly? This needs to be clarified in the text.

There are three measurements shown in the two graphs in Figure 3 (the post-oral arm:mid-body line ratio is repeated in each graph) - none of these measurements is described/mentioned in the figure legend. The composite graphs are confusing as it is unclear from the legend which data points/line represent which measurement on the graph (i.e. which corresponds to the y-axis on the left, and to the y-axis on the right). Legend for the data points would be helpful (see my comments re: all figures above). Also, there is a difference between a ratio (no upper limit) and percentage (ranges 0-100%).

This manuscript primarily tests whether an animal host associates with a phenotype-specific bacterial community, and if the magnitude of morphological change following the expression of plasticity is correlated with the extent that members of the associated bacterial community differ after this trait is expressed.

The three species of sea urchin in this manuscript vary in the magnitude of morphological change exhibited when expressing phenotypic plasticity, with *S. purpuratus* being most plastic, *S. droebachiensis* being least plastic, and *M. franciscanus* being intermediate (see, paragraph 2 of ‘Larval morphometrics’; *S. purpuratus*: 10.9% ($\pm 0.8\%$), *M. franciscanus*: 9.1% ($\pm 1.4\%$), *S. droebachiensis*: 4.5% ($\pm 2.1\%$). To test for a relationship between the magnitude of morphological change and the number of over- and under-represented OTUs post-plasticity (relative to read counts pre-plasticity), we used the “differential_abundance.py” script in QIIME to tally these values. Doing so we specifically compare the number of reads per OTU post-expression of plasticity relative to pre-expression of plasticity (see, Figure 2 A-C).

Figure 3 displays the results of this comparison, with the species of larva being organized from most (left) to least (right) morphologically plastic. This organization was selected based on the hypothesis that taxa that are more morphologically plastic are predicted to better differentially associate with more members of their microbial community than those that are less morphologically plastic. Given this hypothesis, Figure 3 is most accurately organized from either most to least or least to most morphologically plastic, while other arrangements do not allow for this hypothesis to properly be represented.

The organization of Supplemental Figure S7, where we compare to egg size, follows this same logic as Figure 3. The inverse relationship in Supplemental Figure S7 as compared to Figure 3 fits our hypothesis that egg size is inversely related to the extent of morphological plasticity..

22. Supp Figures 7B and 7C: I believe the red bars in 7C depict over-represented OTUs, and the blue bars represent the under-represented OTUs, and that these correspond to the Over:Under ratios shown in 7B? This is unclear from the figure legend/text.

To clarify this, we have added a legend to Supplemental Figure 7C to explain these parts of the figure more clearly.

23. L102-104: Figure 2 with six panels is simply mentioned here once. Figures 2D, 2E and 2F were mentioned later in L119. Should this be a main figure or better presented as a Supplemental Figure? Also, in Figures 2D-2F, what do A and B above the bars denote?

The underlying hypothesis and motivation for this work was to test for convergent shifts in host-associated microbial communities across environmentally elicited phenotypes. Of our six main figures, Figure 2 was the primary figure to support phenotype-specific microbial communities. Complementary to our community-level analysis (Figure 2A-C), we found support for phenotype-specific patterns for specific bacterial groups (Figure 2 D-F, Supplemental Figure 8). Even though Figure 2D-F are mentioned once in the text, we believe these data are an important complement to the community-level analysis to show that phenotype-specific patterns are observed at more than the community-level. Furthermore, inclusion of these taxa may serve as the foundation for directed metagenomic characterization of phenotype-specific microbial communities and of specific taxa.

24. L105-113: The trends described here assume the presentation order of the three species is meaningful (e.g. a time series, or progression of development stages), which is not the case. The text here is misleading and the results are interpreted inadequately. See my comments above about Figure 3.

Inquiries 21 (above) and 24 largely concern the same focus of Figure 3 and its related text. Our response to Inquiry 21 directly apply here. Moreover, we wish to re-emphasize that the organization of these species in Figure 3 is related to the focal life-history character (plasticity) of these species. Specifically, *S. purpuratus* is most plastic (left), *M. franciscanus* is intermediate (middle), and *S. droebachiensis* is least plastic (right). To test for a relationship between the expression of this life-history character and differentially associating with members of the associated microbiome, we test for the correlation between these factors.

25. L116-128: it is unclear why the different bacterial phyla were chosen in Supp Figs 8A-8C for each echinoid species. They represent different resolutions: some are at family level (Bradyrhizobiaceae, Colwelliaceae), some are at genus level (Oleispira, Pseudomonas). Are these the most represented bacterial taxa in each species?

In Figure 2D-F we provide support that along the phenotypic continuum, larvae differentially associate with alpha- and gamma-proteobacteria, with gamma-proteobacteria being over represented when phenotypic plasticity is expressed.

In addition to displaying class-level comparisons, we wished to present data that this same pattern (*i.e.*, phenotype-specific associations) occurred at lower taxonomic levels, including for bacterial families and genera. The only qualification for selecting these bacterial families and genera were phenotype-specific associations. In Supplementary Figure 8 we provide these examples and may suggest different mechanisms of regulation, which we hope to address in future research with directed functional assays.

26. Supp Figs 10, 12 and 14 show distributions of bacterial taxa in each of the three echinoid species before expression of phenotypic plasticity. Those after expression of phenotypic plasticity are shown in Supp Figs 11, 13, 15. It would be of tremendous help if the colour scheme for the different bacterial phyla were consistent across these graphs, especially if we were to assess the before-and-after comparison for a single sea urchin species. The before-and-after snapshot is more interesting than the various PCA plots - perhaps Supp Figs 10 and 11 could be presented as a main figure instead.

While we agree that identical colors for particular bacterial genera across these Supplemental Figures would make it more intuitive for cross-figure comparisons, the color scheme is automatically generated by Prism (the program used for all bar charts). Using the easy-to-use taxonomic tables provided with each bar chart, interested readers can easily compare genera between figures to determine relative conservation of taxonomic groups across species or treatments. More broadly, there are actually few overlapping genera between urchin species, developmental stages, treatment, and time points; thus, for the majority of taxa, each figure would be composed of unique colors and would exhaust the discernable color range rendering comparisons to be difficult.

27. L175: Supp Figs 18-21: similar to my comments above (Supp Figures 10-15) regarding the consistency of colour schemes. The presentation of results appears to have skipped Supp Fig 17, which is presented later in L201-202.

Please see our response to inquiry ‘26’ above. Furthermore, Supplemental Figure 17 is a microbial dendrogram and is directly related to the other Supplemental Figures mentioned by the Reviewer. We have re-numbered this Supplemental Figure, with it now being Supplemental Figure 17.

28. L176: “Supplemental Figure 22”. Also, in the figure legend, did you mean “COI sequences”? “CO1” is a not a standard gene name. In addition, a phylogenetic relationship based on only three sequences/taxa is not meaningful. How was the tree reconstructed? This should be at least mentioned in the text if not in the Methods section.

We thank the reviewer for this correction. We have changed “CO1” to COI” throughout the manuscript. In addition, we agree with Reviewer 3 and recognize that three sequences/taxa are insufficient to make any meaningful claim.

In the revised manuscript, we have removed these data (original ‘Supplemental Figure 22’) from the Supplemental Figures. However, we have kept the statement, “This result mirrors a phyllosymbiotic pattern³² (data not shown), although the number of echinoid species is insufficient for a robust comparison” and replaced “Supplemental Figure 22” with “data not shown.”

29. L232-280: you reviewed the literature the potential gene functions of sea urchins that may play a role in morphological plasticity in diet-restricted environments, and how this may contribute to changes in the associated microbiomes. It remains unclear how this is

relevant to this study, as you did not assess the gene expression of sea urchins under the conditions in your experiment.

We agree with Review 3 that this section of the manuscript is beyond the scope of our data, and we have largely removed this 48-line section. We, however, retained some text from this section on urchin gene expression for a paragraph suggesting that follow-up research should investigate host gene expression and how these patterns may directly relate to the associated microbial community. This reworked section is now paragraph 4 of the Discussion, and begins with “Corresponding with a predicted convergence...”

30. L305-306: the results suggest that microbiome compositions may contribute to phenotypic plasticity in the echinoid larvae, but they do not “support the hypothesis that the evolution of phenotypic plasticity in hologenomic” as the text currently claims. Also, this stated hypothesis remains vague. To really test the hypothesis related to hologenomes, hologenome data (i.e. genes, functions inferred from the genomes of the sea urchins and the microbiomes) are necessary.

We agree with Reviewer 3. To reflect this, we have modified this to read, “Taken together, data presented here support the hypothesis that sea urchin larvae have phenotype-specific microbial community and that the extent to which plasticity of the host can be expressed also includes differential associations with the microbial community.”

31. L327-329: this sentence appears incomplete.

For several decades methods to culture marine invertebrate larvae have used 0.22- μm seawater to exclude “microbial contaminants” (Strathmann, 1987, pg. 4). We modified this “traditional” culturing technique by rearing larvae with 5.0- μm filtered seawater to include the environmental microbes. We have revised this sentence that now reads: “Fertilization of eggs and larval rearing followed Strathmann⁵³, expect, to include the environmental microbiota, embryos and larvae were reared using 5.0- μm filtered seawater (instead of traditional filtration at 0.22- μm)”.

32. L333-334: it is unclear what you mean by “90-95% water changes” every other day. Did you replenish/replace 90-95% of the seawater every other day?

To avoid the build-up of organics and larval waste as well as the depletion of phytoplankton (partially due to larval feeding), we followed common culturing practice (see, citation ‘53’) for 90-95% of the filtered seawater-phytoplankton mix to be removed and then replenished every other day by reverse filtration. This standard procedure was done here to maintain healthy cultures of sea urchin larvae.

33. L368, L377-378: the S in “16S” is always in capital.

We have made this change in the revised manuscript.

34. L373-374: Knowing the length of these MiSeq sequence reads would be helpful. I imagine there is no assembly step involved in this work?

The method applied for this analysis was based on well-established primers designed to amplify V3/V4 for 16S rDNA for bacteria to describe microbial diversity in heterogeneous samples. The product size is approximately 450 bp, which is ideal for maximizing MiSeq read lengths. We verified this amplicon length on agarose gels prior to sequencing. Sequence quality and length were checked with FastQC and verified to be highly similar to agarose gel products.

REVIEWERS' COMMENTS:

Reviewer #2 (Remarks to the Author):

The authors have satisfactorily addressed my comments. I recommend the manuscript for publication.

Reviewer #3 (Remarks to the Author):

The authors have addressed most of my comments satisfactorily. They have toned down their claims on the influence of microbiomes to gene expression in the host, and the claim that "the evolution of phenotypic plasticity is hologenomic".

I list some minor comments below. Some of my earlier points remained unaddressed. Line numbers follow the generated PDF.

1. L24: the results support the hypothesis, not "we" per se.

2. L33: "natural section" remains incorrect. I still believe that you meant to say "natural selection".

3. L88-97: these three sentences remain problematic. For example, "For *S. purpuratus*, larvae fed 100 cells μ mL⁻¹ exhibited morphological plasticity following two versus three or four weeks of diet-restriction (Figure 1A; Supplemental Figure 2A), where the ratio between post-oral arms and larval body increased 10.9% (\pm 0.8%)."

Is the "10.9%" the increase from 2-week to 3-week, or the increase from 2-week to 4-week? The same issue occurs in the following two sentences. It is unclear in each sentence which comparison the presented percentage is referring to.

4. L82-84: while the authors provided a reasonable explanation in the rebuttal, the main text that reads "larvae at the same developmental stage fed the same diet exhibited a higher post-oral arm to mid-body line ratio with time" remains unjustified. This should be clarified in the main text as well, because as Figure 1 is showing, they do not "exhibit a higher post-oral arm to mid-body line ratio with time".

5. Clarity of figure with clear labels in the figure panels: as far as I can tell, the main figures are the same as the original submission. Having the three species names on the figure panels would be very helpful.

6. Order of the three species presented in Figure 3: the authors justified this twice in the rebuttal, but it remains unclear in the main text why the three species are presented in that order. This needs to be made explicit in the main text early on in the Results section.

7. L302-304: the revised sentence appears convoluted. By "expect" did you mean "except"?

Response to Referees:

The Authors thank Reviewers 2 and 3 for reviewing our revised manuscript entitled, “Convergent shifts in host-associated microbial communities across environmentally elicited phenotypes.”

We were pleased that the Reviewers thought our revised manuscript improved based on the comments following the original submission and was worth of publication.

In this second revision we have made the suggested changes by Reviewer 3 as well as those suggested by the associate editor.